# How Income Inequality and Race/Ethnicity Drive Obesity in U.S. Adults: 1999–2016

**DOI:** 10.3390/healthcare9111442

**Published:** 2021-10-26

**Authors:** Hossein Zare, Danielle R. Gilmore, Ciana Creighton, Mojgan Azadi, Darrell J. Gaskin, Roland J. Thorpe

**Affiliations:** 1Department of Health Policy and Management, Johns Hopkins Bloomberg School of Public Health, Baltimore, MD 21205, USA; dgaskin1@jhu.edu; 2Global Health Services and Administration, University of Maryland Global Campus (UMGC), Largo, MD 20774, USA; 3Trachtenberg School of Public Policy and Administration, George Washington University, Washington, DC 20052, USA; danielleg@email.gwu.edu; 4Office of the Deputy Mayor for Health and Human Services, DC Government, Washington, DC 20004, USA; ciana.creighton@gmail.com; 5School of Nursing, Johns Hopkins University, Baltimore, MD 21205, USA; mazadi2@jhmi.edu; 6Health Services Management, University of Maryland Global Campus (UMGC), Adelphi, MD 20774, USA; 7Department of Health, Behavior, and Society, Johns Hopkins Bloomberg School of Public Health, Baltimore, MD 21205, USA; rthorpe@jhu.edu

**Keywords:** income inequality, obesity, Gini coefficient, racial disparities

## Abstract

Obesity is a major public health problem both globally and within the U.S. It varies by multiple factors, including but not limited to income and sex. After controlling for potential covariates, there is little evidence to determine the association between income and obesity and how obesity may be moderated by sex and family income. We examined the association between income and obesity in U.S. adults aged 20 years and older, and tested whether this relationship differs by race or ethnicity groups. For this analysis, we used data from the 1999–2016 National Health and Nutrition Examination Surveys (NHANES). Obesity was determined using Body Mass Index ≥ 30 kg/m^2^; the Gini coefficient (GC) was calculated to measure income inequality using the Poverty Income Ratio (PIR). We categorized the PIR into five quintiles to examine the relationship between income inequality and obesity. For the first set of analyses, we used a modified Poisson regression in a sample of 36,665 adults, with an almost equal number of men and women (women’s ratio was 50.6%), including 17,303 white non-Hispanics (WNH), 7475 black non-Hispanics (BNHs), and 6281 Mexican Americans. The models included age, racial/ethnic groups, marital status, education, health behaviors (smoking and drinking status and physical activities), health insurance coverage, self-reported health, and household structure (live alone and size of household). Adjusting for potential confounders, our findings showed that the association between PIR and obesity was positive and significant more frequently among WNH and BNH in middle and top PIR quintiles than among lower-PIR quintiles; this association was not significant in Mexican Americans (MAs). Results of GC in obese women showed that in comparison with WNHs (GC: 0.34, S.E.: 0.002), BNHs (GC: 0.38, S.E.: 0.004) and MAs (GC: 0.41, S.E.: 0.006) experienced higher income inequality, and that BNH obese men experienced the highest income inequality (GC: 0.45, S.E.: 0.011). The association between PIR and obesity was significant among WNHs and BNHs men in the 3rd, 4th and 5th PIR quintiles. The same association was not found for women. In treating obesity, policymakers should consider not only race/ethnicity and sex, but also strategies to reduce inequality in income.

## 1. Introduction

Is there anything wrong with people getting rich if no one else is harmed in the process? Poverty is concerning, but what is the problem if some people make more money than others? Moreover, why should we be worried about income distribution and income inequality? These are the main questions among mainstream economists; some believe that people should worry less about inequality [1].

Income inequality has sharply increased in recent years. Statistics show that the top one percent of the population is capturing a much larger share of total income growth. In the last four decades, the actual annual earnings for the top 1% and bottom 90% increased 158% and 24%, respectively, with wider gaps between men and women [2,3].

The alarming trends in income inequality raise questions: what is happing if income inequality stays high? Does it harm all communities or only poor people, and who will suffer more? Economists believe that those at the very top level of income can “plunder the poor and middle classes […] by lobbying, by rewriting the rules […] by rewarding and being rewarded by their cronies in business and government” [1,4]. The above statement may be true in the ‘market’ as a ‘general term’, but how does income inequality impact health industries and, most importantly, health outcomes. Angus Deaton—Nobel Prize winner for economics in 2015—has an excellent response to this question. He believes that half of the money spent in the healthcare system in the U.S. has nothing to do with population health. The other half benefits major pharmaceutical and insurance companies, device manufactures, hospitals and physicians; meanwhile, the “richest Americans live in a world in which public goods—including public health care—are irrelevant” [1]. When inequality is large enough, wealthy people live in a very different society from everyone else. It is one of the main reasons why the gap between the richest and poorest threatens communities and public health. Almost all socioeconomic factors, including income, are influenced by race and ethnicity. The National Academies of Sciences, Engineering, and Medicine (2017) report that African Americans and the poor consistently exhibit higher rates of multiple diseases, physical and mental impairment, functional limitations, and disabilities [5]. The combination of low-income communities, minority groups, and high poverty creates an area with the most vulnerable populations; this is the area in which we can expect to find very high adverse health outcomes, including obesity.

### Income Distribution and Obesity

One study has shown additional hazards to the health of people in communities with unequal distribution of income [6]. Studies have also reported a negative association between income inequality and poor health outcomes [7,8]. Income influences health outcomes by decreasing the barriers to accessing care or by moderating environmental factors, living in healthier neighborhoods, or shaping health behaviors [9,10]. Income has direct and indirect effects on health by facilitating an opportunity to control life circumstances and to promote social participation [7].

Obesity has become a national public health concern, with an increasing rate of 42.4% between 1999–2018 [11]. It plays a significant role in several adverse health outcomes, including hypertension, strokes, heart attacks, and diabetes [12]. It is also associated with social, psychological, and economic consequences. For example, Americans pay more than $147 billion to nearly $210 billion per year in the medical costs of adult obesity [13,14]. The concern increases when we include neighborhood and racial/ethnic minorities elements, and minority populations who are more likely to live in areas with the highest poverty rates, lowest financial and environmental resources, and higher rates of single mothers [15]. Our study showed that racial inequality in median income was associated with fewer grocery stores and more fast-food restaurants. As expected, “structural racism on the county-level is associated with obesity and obesogenic environments” [16].

In addition to race/ethnicity and environment, sex also plays a role in obesity. The present study shows that the relationship between income and higher weight status operates differently for men and women [17,18]. It could be because of higher drinking behavior for men [19] or higher exercise activities for women [20], eating more healthy foods, or higher probability of women visiting a health care provider [21,22,23].

Despite good studies on the association between health outcomes and income distribution [19,24,25,26], few studies indicate the impact on health outcomes, especially obesity [19,20]. The present study investigated the relationship between obesity and income among racial/ethnic adults aged 20 and older, and tested whether this relationship differs by sex and race.

## 2. Materials and Methods

We used data from the 1999–2016 National Health and Nutrition Examination Surveys (NHANES) [27]. NHANES is a cross-sectional survey that provides nationally representative estimates of health and nutritional status for the U.S. population, with a response rate of 73.2 between 1999–2016 [28,29], and a multistage probability sampling design that makes the sample representative of each of the four regions of the United States [28]. The original sample between 1996–2016 was 40,917 individuals, with a women’s ratio of 50.6%. For the present study, we included participants who were 20 years old and above. We did not include pregnant women (1667) or missing observations for Poverty Income Ratio (PIR) in the analysis; this yielded an analytic sample of 36,665, including 17,303 white non-Hispanics (WNHs), 7475 black non-Hispanics (BNHs), and 6281 Mexican Americans (MAs).

***Outcome Variable*.** Using Body Mass Index (BMI)—derived by dividing weight in kilograms by height in meters squared (kg/m^2^)—we created a binary variable to identify participants who were obese (if BMI ≥ 30) as the outcome variable [30]. The authors recognize the racial, cultural, and social limitations of BMI calculations. However, for the purposes of the literature and analysis, BMI was used to properly investigate relationships between obesity and income.

***Main Independent Variable***. The primary independent variable of interest was the Poverty Income Ratio—the ratio of family income to the poverty threshold. For the regression models, we defined a categorical variable with five quintiles: lowest quintile (PIR < 0.08), second quintile (0.81–1.36), middle quintile (1.37–2.33), fourth quintile (2.34–4.10), and fifth quintile (4.11–5.00).

We calculated the Gini coefficient (GC) for the second set of analyses to measure income inequality. The GC was defined as A/(A + B); if ‘A’ equals zero, then GC will be zero, which means perfect equality. If ‘B’ was zero, then the GC will be one, which means absolute inequality [31]; the Lorenz curve represents the actual distribution of income in a given society [19]. *A* is the area between the line of perfect equality (45-degree line) and the Lorenz curve; *B* is the area between Lorenz curve *X* and *Y*-axis. The Lorenz curve is a graphical technique to estimate how income is distributed among a population; that is, a higher distance from the line of perfect equality means a smaller percentage of the population receives most of the wealth, and the country’s income distribution is uneven.

***Covariates.*** For the demographic variables, we included age (years) and socioeconomic status (SES) variables. SES was assessed using a binary variable for marital status (1 = married, 0 = otherwise), educational categories (less than high school graduate, high school graduate or general equivalency diploma (GED), more than high school education or some college, or college graduate and above). Health-related characteristics included having health insurance (1 = yes; 0 = no) and self-reported health (excellent–very good, good–fair, and poor). Three variables measured health behavior: smoking (never smoked, a former smoker, or current smoker), drinking (never drink, former drinker, or current drinker), and physical activity (a binary variable showed that an individual had not participated in vigorous activities (1 = yes; 0 = no) during a typical week. The household structure is an essential element in predicting individual/household income [32], so we used a binary variable to present living alone (1 = yes; 0 = no) and number of people in a household.

For the first set of analyses, the mean and proportional differences between WNH, BNH and MA for obesity, demographics, SES, health-related characteristics, and health behaviors were evaluated using an unequal *chi-square test* for categorical variables and *ANOVA* for the continuous variable, e.g., age. In our sample, the prevalence of obesity was greater than 10%; therefore, we used a weighted modified Poisson regression analysis [33,34,35] that produced prevalence ratios (PR) and corresponding 95% confidence intervals (CI) [33,34,35]. We ran sets of weighted modified Poisson regression analyses. Because the interaction between income quintile and race was significant (*p* < 0.001), we stratified the analyses by race. Our findings from a similar study [18] showed a different association between obesity in men and women. For the second set of analyses, we stratified the analyses by sex within each race/ethnic group.

Additionally, to exclude potential endogeneity problems, using family size as an instrumental variable, we ran several sets of IV Poisson regression models as a sensitivity analysis. “IV Poisson estimates the parameters of a Poisson regression model in which some of the covariates are endogenous. The model is also known as an exponential conditional mean model in which some of the covariates are endogenous” [36].

All analyses were weighted using the NHANES individual-level sampling weights for 1999–2016 (8 waves of data) [37] to make our estimates representative of the U.S. civilian non-institutionalized population. We considered *p*-values < 0.05 as statistically significant, and all tests were two-sided. We used STATA statistical software, version 16, to perform all analyses.

## 3. Results

### 3.1. Association between Income-Level and Obesity

***Study population characteristics.***Table 1 compares the distribution of the sample’s characteristics. Overall, the sample age was 47.0 ± 14.0 years, with a slightly more aging WNH population (48.6 ± 11.3). The majority of the study sample had more than a high school education (60.0%), with a higher rate for WNHs (64.1%) and the lowest rate for MAs, of 31% with more than a high school diploma. About 20% of the population were current smokers, with the highest rate for BNHs (26%), and 76% were current drinkers with the highest rate (79.5%) for WNHs. In comparison with MNHs and MAs, BNHs were more likely to be physically inactive (49%); 13.4% of the population lived alone, with the highest rate for BNHs (16%), and the average population in a household was 5.8 people, with the largest household size for MAs. See Table 1 for detailed information and race/ethnicity groups.

### 3.2. A Gap in Income and Highest Income Inequality in Communities of Color

On average, 35% of the population were obese; in comparison, black NHs with 45.5% had higher obesity rates than MA (40%) and white NHs (35%). Interestingly, the distribution on income inequality were different, e.g., WNHs were more likely to be on the fifth quintiles (42%), then BNHs (20%), followed by MAs (12%).

Between 1999–2016, BNH experienced the highest obesity increases: from 18.6% to 45.9%. Mexican Americans had a 19.5%-point increase in obesity from 17.4% to 36.9%, and the lowest increase was for WNH by 11.3 percentage points: from 21.3% to 32.6% in WNH (see Figure 1, panels A and B).

There is a massive gap between the income-to-poverty ratio between 1999–2016. Figure 1 (panels C and D) compares PIR trends between WNH, BNH, and MA women and men. Between 1999–2016, the average PIR was 3.17, 2.10, and 1.80 in WNH, BNH, and MA. To understand more about these differences, we used Lorenz curves and Gini coefficients.

Figure 2—Lorenz curves—shows the Gini coefficient for the income-to-poverty ratio in white NH, black NH, and Mexican American men and women in the US between 1999 and 2016. To plot these curves, we used average GC with jackknife standard errors. This figure compares GC between obese and non-obese women with different skin colors (Figure 2a) and obese and non-obese men with different skin colors (Figure 2b), and obese and non-obese populations by sex.

In Figure 2a panel A, the solid blue line plots the distribution of income in non-obese WNH women; the dash-red line shows the distribution in WNH obese women. The blue line stays over the red line for about 30% of the population and closer to the perfect equality line, which means lesser income inequality within non-obese (GC: 0.269) and greater inequality within obese women (GC: 0.301). Interestingly, there was a different pattern in the BNH population. As presented in panel B, obese BNH women suffered less from income inequality than non-obese (GC: 0.380 vs. 0.392). The red-dash-line stays over the solid blue line for about 50% of the population and closer to perfect equality. Panel C does not show many differences between obese and non-obese MA women (GC: 0.410 vs. 0.395). The last panel (Panel D) compares income inequality between obese WNH, BNH, and MA; overall, BNH women suffered more from obesity than WNH and MA obese women. As presented, BNH women and MA women suffered more from income inequality than WNH. In obese women, WNHs’ lower 25% of the population observed 8.1% of income, BNH followed 6.2% of income, and MA observed 6.4% of income.

Figure 2b compares the GC between obese and non-obese WNH, BNH, and MA men. In panels A, B, and C, we reached the GC between obese and non-obese in WNH, BNH, and MA men. The red-dash-line represents that obese men stay above the solid blue line (non-obese men), meaning lower-income inequities for all groups of income. Despite the difference between racial groups, all obese men suffered less from obesity. For example, the GC moved between 0.247 (S.E.: 0.004) in obese WNH to 0.254 (S.E.: 0.003) in non-obese groups, with similar patterns in BNH and MA. In panel I, we compared the income inequality between WNH, BNH, and MA obese men. Overall, BNH obsess men suffered more from income inequality than WNH and MA, e.g., obese WNHs’ lower 25% of the population observed 9.4% of income, BNH followed 6.9% of income, and MA observed 7.8% of income. See Table 2 for more details on the GC across race/ethnicity and sex.

### 3.3. The Association between Income and Obesity

The association between PIR and obesity in white NH, black NH, and MA is reported in Table 3. The association between income inequality and obesity is a positive and significant association between obesity and income in the middle and top PIR quintiles in WNH and BNH, but not for the MAs. WNH in the 3rd, 4th, and 5th PIR quintiles suffered 22%, 34%, and 29% times more than the population in the first PIR quintile. It is the same picture for the BNH, with 19.0%, 30.0%, and 38% in the third, fourth, and fifth PIR quintiles.

The adjusted models show that in all racial/ethnic groups, the obese population was more women, high school graduates, and former drinkers with poor or fair health. There are some differences between WNH, BNH, and MA. For example, being married has been positively associated with obesity in BNH and MA but not in WNH. Being physically non-active was positively associated with obesity in WNH and MA but not in BNH. Having health insurance coverage also was associated with obesity in MA, but not in WNH and BNH. Finally, obesity was associated positively with the size of a household in WNH.

### 3.4. How Is the Association between Income and Obesity in Men and Women?

The stratified model by sex showed that the association between income and obesity was consistently significant among top PIR quintiles in WNH and BNH men, but not in MA men. For example, WNH men in the 4th (PR: 1.28, CI: 1.08–1.50) and 5th (PR: 1.20, CI: 1.02–1.42) PIR quintiles suffered more from obesity, with a similar pattern for BNH in the 4th (PR: 1.23, CI: 1.03–1.47) and 5th (PR: 1.28, CI: 1.08–1.52) PIR quintiles. The same association was not found for women. See Table 4 for more details.

**Sensitivity analysis results.** We ran several sets of IV Poisson regression models using family size as an instrumental variable. We then tested the exogeneity of the instrumented variable by using the Wald test [38], We reported the results of the Wald test in Appendix A. Our findings have shown that there was no endogeneity between PIR and family size in black NH and Mexican American families, and because there was no endogeneity, we used the standard Poisson regression models. However, for the white NH household, we were not able to reject the null hypothesis of no endogeneity and we ran sets of IV Poisson regression models; because there were no changes in the signs of PIR quintiles, for the original analysis we used the standard Poisson regression models for WNH to make it possible for the audience to compare the results. We reported the results of the IV passion models for WNH in Appendix B.

## 4. Discussion

This study investigated the relationship between obesity and PIR, and how this association changes between men and women of color (white NH, black NH, and Mexican Americans). Our findings showed that higher income in WNH and BNH men was positively associated with a higher prevalence of obesity, but not in MA men. In women, PIR was negatively related to the prevalence of obesity. For example, WNH women, BNH women, and MA women in higher-income groups experienced a lower probability of being obese.

Our literature search showed that despite a fair number of publications regarding income inequality and health [25,39,40,41,42,43], little is known about the impact of income inequality and obesity in men and women, specifically men and women of color. The association between income inequality and racial/ethnicity showed that BNHs and MAs suffered more from obesity than WNHs.

### 4.1. What Is the Impact of Income Inequality, Racial Composition, and Poverty on Health Outcomes?

The importance of income inequalities is not just about its direct or indirect impact on health outcomes. “It may be incompatible with a well-functioning democracy; the rich may write the rules in their favor, and they may work against the public provision of health care or education, for which they pay a large share but have little personal need” [44]. The combination of income inequality and racial composition creates additional complexity.

Studies have shown that racial composition plays a significant role in health outcomes—even more than income inequality. Income inequalities and racial disparities can create psychosocial stress directly harmful to health [24], with more impact from racial composition than income inequality.

The role of poverty is another element in the cycle of the poverty trap. In his paper, Michael Marmot mentioned that “money is vital for those who have insufficient means to ensure social participation as well as adequate housing, nutrition, and clothing” [45]. Studies have reported that minority communities are more likely to have lower incomes, live in areas with higher poverty rates, and neighborhoods with lower resources, including access to healthy foods [46,47,48,49]. Bell et al. reported that racial inequality in median income was associated with fewer grocery stores and more fast-food restaurants, and that county-level racial composition was associated with obesity [16].

A wide range of studies have reported the impact of obesity, health outcomes, poverty [48] and neighborhood [49]. Sometimes socio-environmental variables play a more vital role than racial composition. One study showed that, after controlling for socio-environmental elements, the likelihood of being obese stays the same for black and white men [50].

The complexity of obesity, income inequality [51], and racial composition of populations increases when we add sex elements. Indeed, this study is unique for including all of these elements together. Differences between men and women can be explained by lower physical activities by men or may be because of better intake of food by women. Our findings in this section were similar to published literature. For example, Campbell et al. (2019) reported that income was negatively associated with highly socially integrated men’s weight status, but was positively associated with weight status among lower socially integrated men [17]. Kim et al. (2018) reported that income inequality and lower poverty percentage are significantly associated with lower obesity rates in men [52].

### 4.2. Education and Lifestyle Changes

Improving the health of the U.S. population by encouraging physical activities and promoting nutrition has come to the forefront of public attention. The latest dietary guidelines for Americans have suggested necessary recommendations such as [53]:Following healthy dietary patterns at an early life stage.Cutting back foods high in solid fats, added sugars, and salt.Eating the right number of calories, meeting food group needs, and being physically active.“Customizing nutrient-dense food and beverage choices to reflect personal preferences, cultural traditions, and budgetary considerations” [53].

However, adopting these critical recommendations is not an easy task for some populations. Specifically for low-income people, maintaining a healthy diet requires money and an ease-of-access to healthy foods. Developing programs such as community CSAs—with prioritization of low-income communities—may help ease access to healthy foods with lower cost [54].

### 4.3. Nutrition Assessment

“A nutrition assessment is a process of determining an individual’s nutritional status” [55]; the depth of assessment depends on the individual and on the presenting situation. Low-income populations and people in poverty cannot obtain a variety of foods, but there is a solution. Based on the IRS Schedule H, non-profit hospitals must spend a justifiable amount of tax exemption credits to communities [56]. Under community need assessment programs, non-profit hospitals need to conduct community health needs assessments and provide community benefit contributions (CBC). The IRS allowed non-profit hospitals to spend community benefit contributions for “the cost of staff hours donated by the organization to the community while on the organization’s payroll, indirect cost of space donated to tax-exempt community groups (such as for meetings), and the financial value (generally measured at cost) of donated food, equipment, and supplies” [56,57].

Policymakers and social activists may think about using the non-profit hospitals’ resources to provide life changes education and healthy food to communities, prioritizing low-income populations, pregnant women, elderly, and single-parent households [18]. Policymakers may also encourage big tech companies such as Google, Facebook, etc., and credit card companies and big supermarkets such as Costco, Walmart, etc., to develop a program to monitor and promote a healthy lifestyle.

Most importantly, policymakers may develop policies such as a minimum wage to address fundamental inequality such as income gaps [2] and wage disparities [58].

There are aspects of the study that deserve comment. We used cross-sectional data; therefore, ruling out the possibility of reverse causation is not possible. Evidence shows that extent bias because of reverse causation is mainly indirect [59]. Additionally, the NHANES has some limitations with the income variable and did not report the real income instead of a categorical variable; otherwise, using household income as a continuous variable could give us a better opportunity to find the impact of income differences instead of a proxy variable PIR.

There are also strengths to this study. To our knowledge, it is the first study to examine the relationship between income inequality and obesity between men and women based on their race/ethnicity with a wide range of NHANES data (1999–2016). We used weighted models that make our findings nationally representative estimates, increasing the generalizability of these results. We need to notice that the Lorenz curves are unaffected by the mean of the distribution, and “they cannot be used to rank distributions in terms of social welfare, only in terms of inequality” [32]. Because of NHANES publicly available data, we were not able to control our models for geographical variables such as urban and rural areas and neighborhoods or county level to capture more variances. This is the next step in our study.

## 5. Conclusions

There is a positive association between PIR and obesity for white NH and black NH men. There is no association between PIR and obesity for Mexican American men. Our findings show that income inequality plays different roles between racial/ethnic groups and between men and women. Black NH and Mexican American obese men and women experience higher income inequality than white NH men and women. Treatment of obesity needs multifunctional teams and interventions to reduce infrastructural racism. It also needs improvements in environments and neighborhoods, easy access to healthy foods, behavioral changes by doing more exercise, and targeting the most vulnerable populations. Policymakers may consider providing a monthly allotment of healthy calories instead of a monthly allotment of dollars to low-income populations, and also considering community need assessment analysis.

## Figures and Tables

**Figure 1 healthcare-09-01442-f001:**
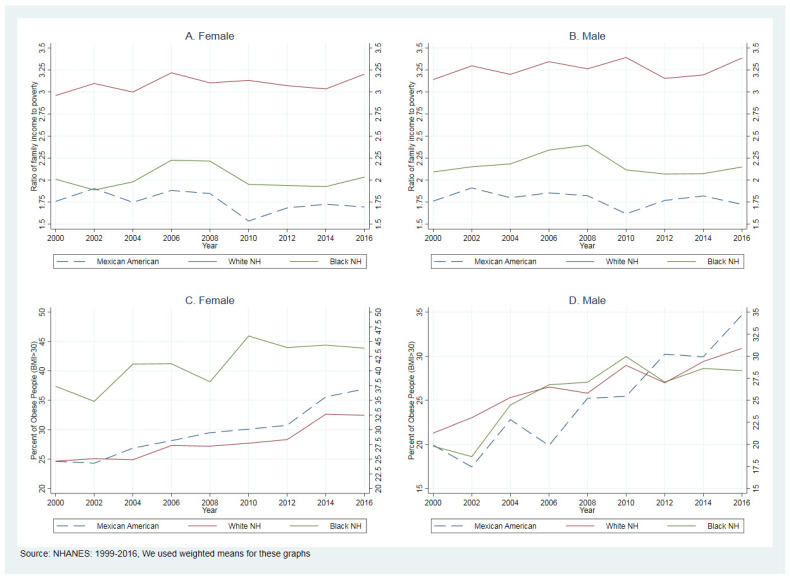
Comparing the ratio of family income to poverty and percent of obese people (BMI > 30).

**Figure 2 healthcare-09-01442-f002:**
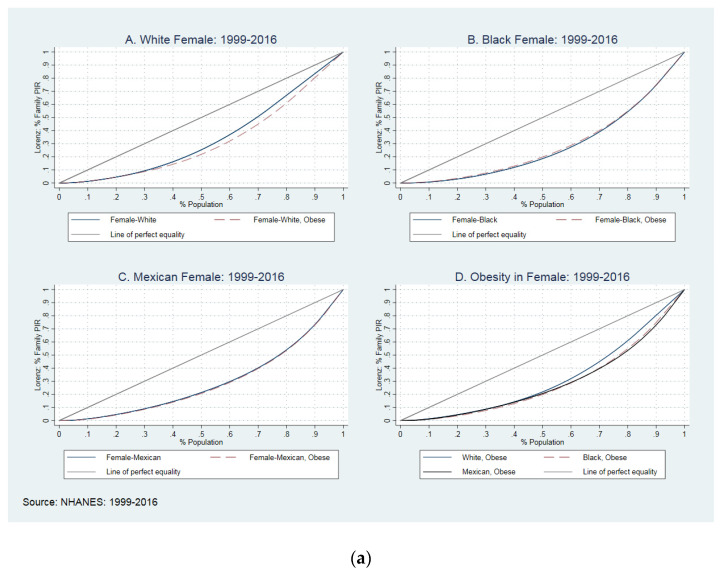
(**a**) The Lorenz curves and Gini coefficients in women, 1999–2016. (**b**) The Lorenz curves and Gini coefficients in men, 1999–2016.

**Table 1 healthcare-09-01442-t001:** Distribution of selected characteristics of U.S. adults over 20 years of age between 1999–2016, National Health and Nutrition Examination Survey (*n* = 36,665).

	All *n* = 36,665	White NH *n* = 17,303	Black NH *n* = 7475	Mexican American *n* = 6281	
	Mean	Mean	Mean	Mean	*p*-Value
Obese (%)	34.9%	33.7%	45.5%	40.0%	0.000
**Ratio of family income to poverty (%)**					
1st quintile (PIR: <0.80)	9.4 (24.0)	6.0 (16.0)	18.0 (44.6)	20.8 (50.8)	
2nd quintile (PIR: 0.81–1.36)	12.9 (27.6)	10.4 (20.5)	17.1 (43.7)	24.2 (53.6)	
3rd quintile (PIR: 1.37–2.33)	17.7 (31.5)	16.2 (24.8)	21.9 (47.9)	23.9 (53.4)	
4th quintile (PIR: 2.34–4.10)	24.6 (35.5)	2.6 (29.3)	23.4 (49.1)	18.9 (49.0)	
5th quintile (PIR: 4.11–5.00)	35.4 (39.4)	41.8 (33.1)	19.6 (46.0)	12.0 (40.7)	
**Socio-demographic variables**					
Age in years at screening (mean, SD)	46.9 (13.8)	48.6 (11.3)	44.5 (18.4)	40.1 (18.1)	0.000
Female sex (%)	50.6%	50.6%	53.8%	44.9%	0.000
**Racial/ethnic groups (%)**					
White NH	70.9%	100.0%	-	-	-
Black NH	10.5%	-	100.0%	-	
Mexican American	7.6%	-	-	100.0%	
**Marital status (%)**					
Married	64.2%	67.1%	44.5%	68.2%	0.000
**Education (%)**					
Less than high school	16.7%	11.4%	24.4%	48.1%	0.000
High school graduate/GED	23.7%	24.5%	25.1%	21.0%	0.432
More than high school	59.6%	64.1%	50.5%	31.0%	0.000
**Health system variables (%)**					
Covered by any kind of health insurance	82.3%	87.4%	76.5%	53.5%	0.000
**Health behaviors**					
**Smoking status (%)**					
Never	52.8%	49.8%	58.7%	61.6%	0.000
Former	25.1%	28.1%	15.4%	19.9%	
Current	22.1%	22.1%	25.9%	18.5%	
**Drinking status (%)**					
Never	12.6%	10.3%	18.5%	15.4%	0.000
Former	11.5%	10.2%	16.6%	12.3%	
Current	76.0%	79.5%	65.0%	72.3%	
**Physical inactivity (%)**					
Has no rigorous or moderate activities	40.5%	37.6%	48.6%	52.6%	0.000
**Self-reported health (%)**					
Fair–poor (=1, if fair–poor)	16.7%	13.6%	23.3%	31.7%	0.000
**HH structure (%)**					
Live alone (=1, if alone)	13.4%	14.7%	15.9%	4.5%	0.094
Total number of people in the household	5.78	5.93	4.53	6.14	0.000

NH = non-Hispanic. Note. We defined quintile based on the ratio of family income to poverty.

**Table 2 healthcare-09-01442-t002:** Comparing Gini coefficient across racial/ethnical groups and sex in U.S. adults in the 1999–2016.

	All *n* = 36,665	White NH *n* = 17,303	Black NH *n* = 7475	Mexican American *n* = 6281	
	Gini Coeff. (SE)	Gini Coeff. (SE)	Gini Coeff. (SE)	Gini Coeff. (SE)	*p*-Value
**Women**					
Obese	0.342 (0.002)	0.301 (0.004)	0.380 (0.004)	0.410 (0.006)	0.000
Non-Obese	0.306 (0.002)	0.269 (0.003)	0.392 (0.005)	0.395 (0.005)	0.000
**Men**					
Obese	0.285 (0.003)	0.247 (0.004)	0.449 (0.011)	0.372 (0.15)	0.000
Non-Obese	0.298 (0.002)	0.254 (0.003)	0.440 (0.004)	0.416 (0.004)	0.000

**Table 3 healthcare-09-01442-t003:** Association between PIR and obesity across racial/ethnic groups in U.S. adults, 1999–2016.

	White NH *n* = 17,303	Black NH *n* = 7475	Mexican American *n* = 6281
**Ratio of family income to poverty (%) (Ref. if PIR < 0.80)**	b/se	b/se	b/se
2nd quintile (PIR: 0.81–1.36)	1.11	0.99	0.98
	[0.91–1.34]	[0.80–1.21]	[0.84–1.14]
3rd quintile (PIR: 1.37–2.33)	1.22 *	1.19 *	1.05
	[1.03–1.45]	[1.00–1.41]	[0.89–1.24]
4th quintile (PIR: 2.34–4.10)	1.34 ***	1.30 **	1.11
	[1.15–1.57]	[1.11–1.54]	[0.93–1.33]
5th quintile (PIR: 4.11–5.00)	1.29 **	1.38 ***	1.01
	[1.10–1.52]	[1.18–1.62]	[0.79–1.29]
**Socio-demographic variables**			
Female	1.36 **	1.83 ***	1.40 ***
	[1.13–1.64]	[1.56–2.14]	[1.22–1.61]
Age	1.00	1.00	1.00 *
	[1.00–1.00]	[1.00–1.00]	[1.00–1.01]
Married	1.05	1.10 **	1.13 **
	[0.98–1.12]	[1.03–1.17]	[1.05–1.21]
**Education (Ref. less than high school)**			
High school graduate/GED	1.14 ***	1.10 **	1.24 ***
	[1.06–1.23]	[1.03–1.17]	[1.13–1.36]
More than high school	1.06	1.09 *	1.25 ***
	[0.98–1.15]	[1.02–1.17]	[1.13–1.38]
**Health system and behaviors**			
Covered by any kind of health insurance	1.08	1.06	1.15 ***
	[1.00–1.18]	[0.99–1.14]	[1.06–1.25]
**Smoking status (Ref. never smoked)**			
Former smoker	1.09 **	1.03	1.08
	[1.03–1.16]	[0.96–1.10]	[0.99–1.19]
Current smoker	0.78 ***	0.83 ***	1.03
	[0.72–0.84]	[0.77–0.89]	[0.93–1.15]
**Drinking status (Ref. never drink)**			
Former drinker	1.24 ***	1.10 *	1.00
	[1.13–1.36]	[1.01–1.20]	[0.88–1.14]
Current drinker	0.95	1.04	0.98
	[0.88–1.03]	[0.96–1.13]	[0.87–1.09]
**Physical activities**			
No rigorous activities	1.37 ***	0.98	1.18 ***
	[1.30–1.44]	[0.92–1.04]	[1.11–1.26]
**Self-reported health: Fair–poor (=1, if fair–poor)**	1.42 ***	1.38 ***	1.35 ***
	[1.34–1.51]	[1.31–1.45]	[1.25–1.45]
**HH structure**			
Live alone (=1, if alone)	1.10	1.04	0.97
	[0.99–1.22]	[0.95–1.13]	[0.81–1.16]
Household’s size	1.06 ***	1.02	1.01
	[1.03–1.08]	[1.00–1.04]	[0.98–1.04]

* *p* < 0.05, ** *p* < 0.01, *** *p* < 0.001. PR = prevalence ratio. Notes: We have defined quintile based on the GC calculated from ratio of family income to poverty.

**Table 4 healthcare-09-01442-t004:** Association between income differences and obesity across racial/ethnic groups in U.S. adults, 1999–2016.

	White NH N = 17,303		Black NH N = 7475		Mexican American N = 6281	
	Male	Female	Male	Female	Male	Female
**Ratio of family income to poverty (%) (Ref. if PIR < 0.80)**	PR, 95% CI	PR, 95% CI	PR, 95% CI	PR, 95% CI	PR, 95% CI	PR, 95% CI
2nd quintile (PIR: 0.81–1.36)	1.10	1.01	0.96	1.02	0.97	0.93
	[0.90–1.33]	[0.89–1.14]	[0.78–1.19]	[0.92–1.14]	[0.83–1.13]	[0.82–1.07]
3rd quintile (PIR: 1.37–2.33)	1.19	1.05	1.15	1.01	1.01	0.90
	[1.00–1.41]	[0.92–1.19]	[0.97–1.37]	[0.91–1.14]	[0.86–1.20]	[0.78–1.05]
4th quintile (PIR: 2.34–4.10)	1.28 **	1.01	1.23 *	1.02	1.04	0.91
	[1.08–1.50]	[0.89–1.15]	[1.03–1.47]	[0.89–1.16]	[0.86–1.26]	[0.73–1.12]
5th quintile (PIR: 4.11–5.00)	1.20 *	0.88	1.28 **	0.98	0.93	0.89
	[1.02–1.42]	[0.76–1.01]	[1.08–1.52]	[0.87–1.10]	[0.72–1.20]	[0.73–1.09]
**Socio-demographic variables**						
Age	1.00	1.00	1.00	1.00	1.00	1.00 **
	[1.00–1.00]	[1.00–1.00]	[1.00–1.00]	[1.00–1.00]	[1.00–1.00]	[1.00–1.01]
Married	1.10	1.01	1.34***	1.00	1.30***	1.03
	[1.00–1.21]	[0.92–1.10]	[1.19–1.52]	[0.93–1.07]	[1.15–1.47]	[0.92–1.14]
**Education (Ref. less than high school)**						
High school graduate/GED	1.15 *	1.13 *	1.28 ***	1.00	1.32 ***	1.14
	[1.01–1.31]	[1.03–1.25]	[1.14–1.43]	[0.91–1.11]	[1.15–1.51]	[1.00–1.30]
More than high school	1.13	1.01	1.20 *	1.03	1.30 ***	1.20 **
	[0.99–1.28]	[0.91–1.12]	[1.04–1.38]	[0.94–1.12]	[1.13–1.49]	[1.06–1.36]
**Health system and behaviors**						
Covered by any kind of health insurance	1.17 *	1.00	1.05	1.05	1.26 ***	1.04
	[1.03–1.31]	[0.90–1.11]	[0.93–1.19]	[0.96–1.14]	[1.11–1.44]	[0.93–1.15]
**Smoking status (Ref. never smoked)**						
Former smoker	1.12 **	1.06	1.00	1.04	1.04	1.13
	[1.03–1.21]	[0.97–1.15]	[0.89–1.13]	[0.95–1.12]	[0.92–1.19]	[0.98–1.30]
Current smoker	0.76 ***	0.79 ***	0.74 ***	0.90 *	0.95	1.19 *
	[0.68–0.85]	[0.71–0.88]	[0.65–0.84]	[0.83–0.98]	[0.83–1.08]	[1.04–1.37]
**Drinking status (Ref. never drink)**						
Former drinker	1.34 ***	1.21 ***	1.12	1.09	0.81	1.05
	[1.15–1.55]	[1.09–1.34]	[0.92–1.36]	[1.00–1.19]	[0.62–1.07]	[0.92–1.21]
Current drinker	1.02	0.93	1.08	1.02	0.90	0.98
	[0.90–1.16]	[0.85–1.02]	[0.92–1.26]	[0.94–1.11]	[0.73–1.11]	[0.86–1.11]
**Physical activities**						
No rigorous activities	1.33 ***	1.40 ***	0.99	0.96	1.20 ***	1.15 **
	[1.23–1.43]	[1.31–1.50]	[0.89–1.09]	[0.90–1.03]	[1.09–1.32]	[1.05–1.27]

**Self-reported health: Fair–poor (=1, if fair–poor)**	1.38 ***	1.46 ***	1.45 ***	1.34 ***	1.29 ***	1.38 ***
	[1.25–1.51]	[1.36–1.58]	[1.31–1.61]	[1.26–1.43]	[1.15–1.45]	[1.27–1.51]
**HH structure**						
Live alone (=1, if alone)	1.16 *	1.05	1.26 **	0.95	1.14	0.88
	[1.01–1.34]	[0.92–1.20]	[1.06–1.49]	[0.85–1.05]	[0.86–1.50]	[0.72–1.07]
Household’s size	1.07 ***	1.04 **	1.03	1.01	1.01	1.01
	[1.03–1.10]	[1.02–1.07]	[1.00–1.07]	[0.99–1.03]	[0.97–1.05]	[0.97–1.04]

* *p* < 0.05, ** *p* < 0.01, *** *p* < 0.001. PR = prevalence ratio.

## Data Availability

The data presented in this study are openly available in the National Health and Nutrition Examination Survey (NHANES) at https://www.cdc.gov/nchs/nhanes/index.htm (accessed on 10 March 2020).

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
