# Peer review of "How Income Inequality and Race/Ethnicity Drive Obesity in U.S. Adults: 1999–2016"

_healthcare, 2021, doi:10.3390/healthcare9111442_

Round 1

Reviewer 1 Report

My main concern lies with the identification strategy. As the authors correctly said, they evaluate the correlation between the Poverty Income Ration and obesity. But in order to provide policy recommendations, we have to exclude endogeneity problems. The authors may use instrumental variables. For example,  family income can be instrumented by the family size. Family size is correlated with family income but not associated with error terms; therefore, it satisfies the conditions for the instrumental variable. 

Reviewer 2 Report

I feel this is a topic of interest and importance.  The study itself is very thorough and methodolgically sound.  However, there is a political thread woven throughout the paper, especially in the introduction.  I suggest the authors focus on the study and the results without editorializing.  I think there are also alternatives to consider in terms of solving the issue at hand, such as giving lower income populations tools such as nutrition education and lifestyle changes.  I think another interesting idea would be a program whereby low income populations are not given a monthly allotment of dollars, but a monthly allotment of calories.  Given the ubiquity of technologies and big data it is not inconceivable that such a program could be developed, assuming of course, careful development and oversight.  I offer these comments without prejudice and am grateful for the dedicated researchers work.  Thank you.

Round 2

Reviewer 2 Report

I find the revised version of the article to be very positive and truly makes for a very interesting and important topic for our readers.  Congratulations on this excellent work.